# Forensic Microbiology: When, Where and How

**DOI:** 10.3390/microorganisms12050988

**Published:** 2024-05-14

**Authors:** Riccardo Nodari, Milena Arghittu, Paolo Bailo, Cristina Cattaneo, Roberta Creti, Francesco D’Aleo, Veroniek Saegeman, Lorenzo Franceschetti, Stefano Novati, Amparo Fernández-Rodríguez, Andrea Verzeletti, Claudio Farina, Claudio Bandi

**Affiliations:** 1Department of Pharmacological and Biomolecular Sciences (DiSFeB), University of Milan, 20133 Milan, Italy; 2Analysis Laboratory, ASST Melegnano e Martesana, 20077 Vizzolo Predabissi, Italy; 3Section of Legal Medicine, School of Law, University of Camerino, 62032 Camerino, Italy; 4LABANOF, Laboratory of Forensic Anthropology and Odontology, Section of Forensic Medicine, Department of Biomedical Sciences for Health, University of Milan, 20133 Milan, Italy; 5Antibiotic Resistance and Special Pathogens Unit, Department of Infectious Diseases, Istituto Superiore di Sanità, 00161 Rome, Italy; 6Microbiology and Virology Laboratory, GOM—Grande Ospedale Metropolitano, 89124 Reggio Calabria, Italy; 7Microbiology and Infection Control, Vitaz Hospital, 9100 Sint-Niklaas, Belgium; 8Department of Infectious Diseases, Fondazione IRCCS Policlinico San Matteo, University of Pavia, 27100 Pavia, Italy; 9Microbiology Department, Biology Service, Instituto Nacional de Toxicología y Ciencias Forenses, 41009 Madrid, Spain; 10Department of Medical and Surgical Specialties, Radiological Sciences and Public Health University of Brescia, 25123 Brescia, Italy; 11Microbiology and Virology Laboratory, ASST Papa Giovanni XXIII, 24127 Bergamo, Italy; 12Romeo ed Enrica Invernizzi Paediatric Research Centre, Department of Biosciences, University of Milan, 20133 Milan, Italy

**Keywords:** forensic microbiology, post-mortem, criminalistics, microbiome

## Abstract

Forensic microbiology is a relatively new discipline, born in part thanks to the development of advanced methodologies for the detection, identification and characterization of microorganisms, and also in relation to the growing impact of infectious diseases of iatrogenic origin. Indeed, the increased application of medical practices, such as transplants, which require immunosuppressive treatments, and the growing demand for prosthetic installations, associated with an increasing threat of antimicrobial resistance, have led to a rise in the number of infections of iatrogenic origin, which entails important medico-legal issues. On the other hand, the possibility of detecting minimal amounts of microorganisms, even in the form of residual traces (e.g., their nucleic acids), and of obtaining gene and genomic sequences at contained costs, has made it possible to ask new questions of whether cases of death or illness might have a microbiological origin, with the possibility of also tracing the origin of the microorganisms involved and reconstructing the chain of contagion. In addition to the more obvious applications, such as those mentioned above related to the origin of iatrogenic infections, or to possible cases of infections not properly diagnosed and treated, a less obvious application of forensic microbiology concerns its use in cases of violence or violent death, where the characterization of the microorganisms can contribute to the reconstruction of the case. Finally, paleomicrobiology, e.g., the reconstruction and characterization of microorganisms in historical or even archaeological remnants, can be considered as a sister discipline of forensic microbiology. In this article, we will review these different aspects and applications of forensic microbiology.

## 1. Introduction

Till the beginning of the XXI century, microbiology has played a relatively minor role in forensic sciences. In the early 1990s, the sequencing of amplified viral genome fragments allowed for the reconstruction of the role of a dentist in the transmission of HIV to several patients in Florida, opening the way toward the application of PCR-based genome typing in forensic microbiology [1]. Since then, the traditional perspective of the clinical microbiologist, called upon to diagnose infections, has over time been enriched with other not strictly clinical domains such as applications useful in a medico-legal context: the diagnosis of hospital-associated infections, sudden death (SD) in adulthood, pediatric and prenatal ages, sexually transmitted infections, the complications of sexual assaults, and forensic microbiome applications. In addition, paleomicrobiology could be regarded as a sister field of forensic microbiology [2].

Forensic microbiology is a novel field of research relying on traditional, culture-based microbiology and molecular analyses carried out on samples collected from the body of the deceased, the infected person or the surrounding environment, providing evidence of sufficient quality to support legal proceedings. Forensic microbiology requires standardization and validation of procedures, from the collection of biological samples from both living and deceased subjects, a robust set of technologies and expertise when interpreting the results [3].

A network of forensic microbiologists, pathologists, geneticists and physicians from different European countries have shaped a flexible protocol providing minimal requirements for post-mortem microbiology (PMM) sampling in cases of SD, bioterrorism, tissue and cell transplantation and paleomicrobiology [4]. Later, the ESCMID Study Group for Forensic and Post-mortem Microbiology (ESGFOR) yielded additional protocols for different scenarios of SD and the PMM in the hospital setting [5].

Mortality rates related to infectious diseases have decreased over the years thanks to the implementation of health policies regarding a better use of hygiene standards, the introduction of new vaccines along with the promotion of vaccination campaigns, the use of antimicrobials and the advances in supportive care for critically ill patients [6]. However, infectious outbreaks in hospitals, bioterrorism as well as epidemic and pandemic events pose real threats to public health. In these cases, the application of forensic microbiology has a pivotal role in identifying the etiological agent, the source of the infection, the mode of transmission and directing containment actions such as the implementation of infection control and prevention programs [7].

In this sense, forensic microbiology and molecular epidemiology in public health share the scope to reconstruct the source of an unexpected infectious fatal event and to apply all possible means for its containment and prevention of further transmission. However, they differ in the rigor of accredited standard operating procedures and dedicated laboratories [8,9].

Forensic microbiology has also gained particular interest in providing new insights in deciphering historical fatal events such as waves of plague and other infectious pestilences or sudden and suspicious deaths of historical figures [10,11].

In this review, we will focus on a series of topics that are relevant to forensic microbiology, from the dynamics of microbial community composition after death to the methods for sampling, and the application of high-throughput tools, such as next-generation sequencing. In addition, we will focus on specific topics, including paleomicrobiology, violent and sudden deaths.

## 2. Patterns of Microbial Dynamics after Death

The microbial population that accompanies a host after its death is referred to as the “Thanatomicrobiome” [12]. This microbial community plays a crucial role in the decomposition process, characterizing the microbial succession in and around the dead biomass [13]. Furthermore, various authors have differentiated between the microbial communities present within the internal organs of a deceased body (known as the thanatomicrobiome) and the microbial communities found on the skin, surface epithelia and external mucous membranes of the corpse (known as epinecrotic microbial communities) [14,15,16].

Although the human microbiome includes bacteria, fungi, viruses and other single-celled organisms, bacteria are the most significant in forensic medicine because of their diversity and primary involvement in decomposition [17]. However, as decay continues, fungi become more and more important in the process [18]. In general, the bacterial sequences found during the colonization of a cadaver are associated with the post-mortem interval (PMI). This process typically begins with the presence of *Staphylococcus* spp., followed by coliforms and *Candida* spp. and eventually concludes with the appearance of mixed populations of anaerobic bacteria [19]. In this context, Metcalf et al. [20] showed that the use of microbial succession during the ecological process of decomposition is a promising technique to estimate the PMI. However, developing and transitioning new forensic science technologies into the justice system requires overcoming scientific, investigative and legal hurdles.

The initial and subsequent composition of the thanatomicrobiome is influenced by the human microbial community present at the time of death, which may vary due to multiple factors, including the personal characteristics of the subject such as age, sex, ethnicity and geographical origin [21]. Other influencing factors can be divided into biotic and abiotic ones. The former include antemortem infections, insects, scavengers and commensal microbial populations. Abiotic factors include environmental conditions to which the corpse is exposed (temperature, humidity, pH), time since death and living habits (e.g., eating habits, the weight of the subject, antimicrobial/drug therapy) [22,23]. All of the above factors can obviously have an impact on PMI estimation.

These factors also influence the speed of the cadaver decay process. For example, open wounds and breached skin can cause faster decay. Women with more subcutaneous adipose tissue decay faster due to increased water content in the interstitium/intercellular space. Slim bodies, malnutrition and poisoning lead to a slower decay [24].

The body’s internal organs, such as the heart, liver, spleen and brain, are sterile during life. After the heart stops beating, tissues begin to deteriorate due to a lack of oxygen. Cellular functions continue until all the oxygen is used and carbon dioxide can no longer be removed. This causes a hypoxic and acidic environment that leads to cell rupture. Enzymes like lipases leak out and promote autolysis, triggering microbial processes responsible for tissue breakdown or putrefaction. During this process, the immune response rapidly decays, cellular junctions become loose and nutrients become available to the microbiota [18,25].

Thus, when the physical and chemical barriers that impede/regulate the movement of bacteria weaken, it becomes easier for bacteria to invade the neighboring tissues. The progression of intestinal microbiota from the gut to the contiguous organs, tissues and body cavities has been studied using mouse models: these suggest that the first step of microbiota progression involves the mesenteric lymph nodes and then extends to other tissues/organs [26,27]. Progression also occurs from the upper airways to contiguous organs [24]. The uterus and prostate are the last internal organs to decompose [27].

Microbial communities utilize released nutrients, leading to the putrefaction stage of body decomposition. One visible sign is the discoloration of sulfhemoglobin due to hydrogen sulfide reacting with blood hemoglobin. This gas is released from a sulfur-containing amino acid breakdown in the cecum, diffusing into surrounding tissues. Abdominal inflation due to gas production and accumulation is a preliminary sign of progression to the bloating stage. If the pressure is severe enough, it can result in post-mortem injury and a change in the microbial environment from anaerobic to aerobic [28,29]. The deflation of the body initiates the onset of active decay, facilitated by microorganisms and scavenging activity until only the skeleton, skin and hair remain [24].

Several studies have investigated which microbial communities were prevalent in specific organs during decomposition. The most abundant taxa of microbial communities according to the organ examined are *Firmicutes* in the brain, heart and spleen [12,24,30]; Firmicutes and Proteobacteria in the liver [12,13,30]; Bacteroidetes, Firmicutes and Proteobacteria in the oral cavity [31]; Acidobacteria, Actinobacteria, Bacteroidetes, Chloroflexi, Firmicutes and Proteobacteria in the bones [32]. According to another study, the predominant phyla in the post-mortem microbiota are Firmicutes and Proteobacteria [33].

Considering the PMI, the human microbiota remains mostly stable during the initial 24–48 h after death. The microbial communities in this timeframe may indicate the individual’s health condition and dysbiosis before death [34]. Unique microbial communities can be distinguished across various body sites during this period [34,35]. However, environmental changes after this timeframe lead to a microbial succession that transforms the human body and microbiota in a manner that deviates significantly from a living individual [18].

As the period following death becomes longer, the alpha diversity of communities typically declines, as anticipated with nutrient influxes, and the composition of communities, referred to as beta diversity, becomes more alike across bodily locations [36,37].

For instance, various studies have analyzed the microbial populations in both the oral cavity and the intestine at different stages of cadaveric decay. Typical oral microorganisms were detected during the early stages in the oral cavity. However, during the emphysematous phase, bacteria characteristic of both the oral cavity (*Peptostreptococcaceae* and *Bacteroidaceae*) and the intestinal tract (Enterococcaceae) were found [31]. Over time, specific bacterial strains such as *Bacteroides*, *Parabacteroides*, and *Lactobacillus* decreased in the intestine, while taxa belonging to the Enterococcaceae family and *Clostridium* species increased [38,39] and subsequently spread to other organs of the body [40,41]. The adjacent environment impacts the arrangement and development of microbial communities on the skin following death, including the soil’s microbial composition [36].

Other studies have investigated the disparities in the thanatomicrobiome relating to gender. In female cadavers, there was a prevalence of *Pseudomonas* and *Clostridiales*, while male cadavers showed a high abundance of species of *Clostridium*, *Clostridiales* and *Streptococcus*. Specifically, *Pseudomonas* was identified as the most common genus in women, whereas *Rothia* was exclusive to men [42]. Furthermore, when examining cardiac tissues, the genera *Streptococcus* and *Lactobacillus* were exclusively present in men, while women had a higher occurrence of bacteria belonging to *Pseudomonas* and *Clostridium* genera [43]. The type of microbial communities can be affected by insects that feed on decaying flesh. Such insects change the microbiome through their secretions and excretions, including symbiotic microbes and microbial and immune products. One study showed that the presence of bacteria of the genus *Ignatzschineria* marks the presence of insects [44].

During the advanced decomposition stage, microorganisms that are typically found in the soil are present (Alcaligenaceae, Planococcaceae, Pseudomonadaceae [31]; *Acinetobacter*, *Anarosphaera*, *Clostridium*, *Ignatzshineria*, *Peptostreptococcus*, *Providencia*, *Wohlfahrtiimonas* [38]). This is because when a corpse decomposes in soil surrounded by vegetation, the soil’s carbon and nutrient levels increase, creating ideal conditions for these microorganisms to thrive. Conversely, the dry remains stage is characterized by the predominance of spore-forming microorganisms (*Bacillus*, *Clostridium*). The presence of spores allows these microorganisms to survive even without nutrients [21].

In general, in view of the potential application of forensic microbiology (Table 1), the issues related with the overall process of microbial dynamics after death, and along the chain of custody of corpses/samples, are to be carefully considered. In particular, quality criteria aimed at the sampling, packaging, transport and preserving of samples are a key to success in performing forensic analyses [45]. Further studies assessing the preservation of the microbiomes before and during the analysis of “microbe-based legal evidences” are required in order to develop standard operative procedures of application in forensic science.

## 3. Microbiological Sampling after Death: When, Where and How; Pitfalls and Precautions

Despite significant advances in the diagnosis and treatment of infectious diseases during the past century, microbial infections continue to be a significant cause of mortality worldwide. Traditionally, the relevance of post-mortem microbiologic examinations has been a subject of debate. The problems of implementing routine procedures in daily autopsy practice clearly relate to the lack of consensus on their broader utility as well as to a lack of regulation [66]. However, during the last 10 years, the ESGFOR has made a great effort to yield sampling recommendations [5] and to show the results of application of PMM [67]. 

Before the start of the autopsy, the skin should be disinfected with disinfectants such as 0.05–2% chlorhexidine solution and 0.5% tetrimony bromide solution. If isopropyl alcohol is used, the toxicologist should be informed as it may be used as an internal control for toxicological analysis [5,68]. After opening the thoracic and abdominal cavity, the surfaces of the organs to be sampled should be cauterized with a hot spatula before handling the internal organs and ligating the blood vessels. The area should be in the following order of preference: biopsied or punctured, with a needle and syringe inserted for aspiration, or through a sterile swab sample. 

If a CNS infection is suspected, it is possible to attempt liquor sampling by lumbar puncture before starting the autopsy or, once the skull is open, trying to expose the lateral ventricles or the subarachnoid cisterns and withdrawing the liquor aseptically.

Figure 1 shows a flow chart representing the procedures to be implemented in the event of a death, for autopsy sampling, devoted to the culturing of microorganisms, or other methods for microbial detection and identification.

As microbiological samples should be taken as soon as possible after death, autopsies should be performed within 24 h of death. This can be a problem in some countries as the legal system (or the Prosecutor’s order) may impose longer times for performing the autopsy; in any case, the cadaver should be placed in a sealed body bag at 4 °C as soon as possible before the autopsy begins [66,69].

Classical methods using minimally invasive techniques require the cauterization of all organ surfaces with a hot spatula or soldering iron prior to sampling. Tissue samples must be taken with the organs intact and before extraction of the bowel. This is because handling of the bowel prior to removal increases the amount of contamination in post-mortem cultures due to passive circulation of blood from the contaminated area. Blood, body fluids and nasopharyngeal secretions are best collected at the beginning of the autopsy. Blood cultures obtained from heart blood, spleen cultures or peripheral venous sites are commonly applied during post-mortem examinations; such cultures have a reported positivity rate between 7% and 69% [66,68,69].

There is increasing demand for the development of minimally invasive autopsy (MIA) techniques in fetal, pediatric and adult autopsies; in an MIA, samples of organs, tissues and body fluids are obtained by puncture or needle biopsy without opening the abdomen. In most cases, MIA is performed by blind biopsy using a cutting needle. However, MIA may also be performed by tissue biopsy and fluid removal using image-guided biopsy/needle biopsy [68,70].

All specimens should reach the laboratory within 2 h if stored at room temperature or within 48 h at temperatures between 2 and 8 °C if stored in an appropriate transport medium [66].

Careful efforts should be made to avoid contamination of microbiological samples taken at necropsy. This is because contamination may either be related to external factors or factors associated with the cadaver. To reduce contamination, autopsy rooms and instruments should be easily cleaned and disinfected. It is advisable to sterilize the tools used, also using a Bunsen burner. If possible, replacing the tools for each incision and each sample collection, or using disposable instruments, is advisable. Limiting the use of disinfectant solutions is recommended, as mentioned before, in situations in which toxicological analyses are also planned. Controlled air circulation, tightly closed doors and restricted access, well-trained forensic technicians and the use of personal protective equipment are also essential [71].

Human remains are a complex microbial ecosystem and vary considerably from person to person due to several factors. Therefore, it is important to emphasize that positive results obtained from post-mortem specimens may reflect actual infection but may also reflect contamination of the specimen. In conventional autopsies, the incidence of post-mortem contamination is estimated to be around 20% [68,69]. In this context, ‘contamination’ means the growth of microorganisms in post-mortem cultures that do not harbor the original infectious agent. In many practical cases, a real pathogen can easily be differentiated from contaminants, but interpretation criteria are still to be established to help in routine PMM. 

We emphasize that knowledge of the possible persistence of pathogens in deceased individuals is important to know for the safety of the operators during the autopsy practice, especially when referring to pathogens like *Neisseria meningitidis*, *Mycobacterium tuberculosis* complex, HCV, HIV and emerging pathogens, such as SARS-CoV-2 [72,73].

## 4. Next-Generation Sequencing (NGS) for Microbiological Post-Mortem Analysis

Next-Generation Sequencing (NGS) refers to a set of high-throughput DNA sequencing technologies that have revolutionized genomic research since the end of the 1990s. These technologies allow for the rapid and cost-effective generation of large amounts of DNA sequence data, enabling the analysis of entire genomes, transcriptomes, epigenomes and even mixtures of millions of DNA molecules [74]. This ability to sequence mixtures of DNA molecules has generated new opportunities for researchers to gain insight into the composition and function of microbial communities, with important implications also for post-mortem analysis and forensic microbiology [75,76].

There are several types of NGS technologies available on the market, each with its own advantages and limitations. A brief description of the most used ones follows below.

Illumina NGS is an established and widely used technology that offers high sequencing accuracy and quality, low error rates and high throughput [77]. It has a large community of users and a range of sequencing platforms and applications for different research needs [78,79]. However, Illumina NGS produces short read lengths, which can limit the ability to sequence and assemble complex genomes or identify structural variations [80]. Additionally, library preparation can be time-consuming and labor-intensive, and there may be biases towards sequencing certain regions or types of DNA, as repetitive regions [79,81].

Ion Torrent sequencing is known for its relatively fast speed and low capital costs [82,83]. However, it is also more error-prone, has a shorter read length, lower throughput and a limited range of applications compared to Illumina sequencing [84].

PacBio sequencing is known for its long read length, which can improve genome assembly and enable the identification of structural variations [85], and lack of PCR amplification, which can reduce the risk of amplification bias and other artifacts [86,87,88]. However, when compared to Illumina sequencing, it is generally more expensive, has a lower throughput and can have a higher error rate [89].

Oxford Nanopore is another sequencing technology that allows the generation of long reads [90]. It has portable (sequencers are relatively small) and real-time sequencing capabilities, allowing for rapid detection and analysis of genetic material [90], with potential for field-based applications. It is also able to sequence RNA and other types of nucleic acids [91]. Like PacBio, it does not require PCR amplification [90]. However, Oxford Nanopore technology has lower sequencing accuracy and higher error rates compared to Illumina [92], although the accuracy has been improving with newer versions of the technology [93]. Oxford Nanopore also has limited throughput and higher costs per base compared to Illumina [94].

Compared with traditional bacterial culture and PCR methods, NGS has obvious technical advantages including the reduced cost and time needed to perform a wide-screen molecular analysis on multiple types of samples. These advantages offer an easy and cost-effective way to answer multiple crucial questions in the forensic context.

NGS is incredibly useful in determining the microbial diversity in a specific sample. Targeted sequencing is a method used to selectively sequence specific phylogenetically and/or taxonomically informative gene markers (e.g., 16S rRNA for bacteria and archaea, 18S rRNA for eukaryotes, and internal transcribed spacer—ITS, for fungi) [95] within a genome or set of genomes, rather than sequencing the entire genome. 

In microbiology, the 16S rRNA gene is one of the most frequently used targets, since it allows rapid screening for the presence of a large range of bacterial diversity [96]. NGS 16S sequencing is a high-throughput method that can sequence millions of 16S rRNA gene fragments in a single run, providing a detailed picture of the bacterial community in a given sample. This method uses PCR amplification of the hypervariable regions of the 16S rRNA gene, followed by sequencing of the amplified fragments using NGS platforms. This approach is relatively easy and cheap with respect to shotgun metagenomics (see below), but currently there is no 16S rRNA gene PCR primer pair that is truly ‘universal’ and some species, or even phylum, can evade detection through this method [97].

On the other hand, shotgun metagenomics is a technique used to analyze the microbial community present in a sample by sequencing all of the genetic material present in it [81]. This approach involves randomly breaking apart all the DNA in the sample, sequencing the resulting fragments and then using computational methods to identify and assemble the DNA sequences into individual genomes or microbial community profiles [97]. Shotgun sequencing accounts for higher taxonomic resolution and less bias and provides a greater level of diversity, but the cost is higher than that of targeted sequencing, and datasets can be overwhelmed by the most abundant taxa [98,99].

The study of post-mortem microbial diversity using targeted gene sequencing or shotgun metagenomics can be used to study decomposition processes [38,100,101], estimate the time since death [22,42,99], determine the cause of death [102], or define the microbial signature of a sample/individual which can be used to identify possible links among people or geographic locations [103,104].

NGS can also be used to reconstruct the entire genome of a microorganism. This is extremely useful in hospital settings, or more in general, during infectious disease outbreaks. Determining which specific strain is responsible for the death of a patient/individual can help mapping the spread of the pathogen during an outbreak by identifying potential sources and drug susceptibility [105,106,107]. Detecting, identifying and understanding the potential transmission pathways of a particular pathogen are crucial in allocating resources for infection prevention and control. Additionally, in the event of a legal case, the genotyping data collected can serve as evidence to either confirm or exonerate the hospital from being the source of the infection [53].

The analysis of next-generation sequencing (NGS) data presents a unique set of challenges in forensic investigations. Contamination is a primary concern, since even small traces coming from environmental sources or handling can obscure exogenous microbial signals [108,109]. Mitigating contamination requires meticulous sample collection, processing and stringent bioinformatic protocols to differentiate true microbial signatures from background noise [110]. Additionally, forensic samples often exhibit low biomass, making it challenging to obtain sufficient genetic material for analysis. Moreover, degraded DNA further complicates analysis, as fragmented or damaged sequences can hinder accurate microbial identification and strain typing [111]. Addressing these challenges requires advanced bioinformatic techniques tailored to handle difficult data (low biomass and degraded DNA), implement robust quality control measures and employ sophisticated algorithms capable of accurately detecting and differentiating relevant genetic signals from background noise. Additionally, integration with metadata and contextual information is crucial for proper sample classification and forensic interpretation, ultimately ensuring the reliability and validity of forensic microbiology analyses [112]. Indeed, the challenges encountered in forensic microbiology, such as contamination, low biomass and degraded DNA, are also prevalent in paleomicrobiology [113,114]. Consequently, advancements in one field often benefit the other, fostering a symbiotic relationship where techniques and protocols developed for forensic microbiology can be adapted and utilized in paleomicrobiology, and vice versa, facilitating progress and innovation in both disciplines [2,114].

NGS has also revolutionized transcriptome analysis, enabling high-resolution studies and expanding the scope of forensic applications [115]. Transcriptomics has emerged as a powerful tool in forensic microbiology, offering insights into gene expression patterns that can aid in various aspects of forensic investigation. From identifying body fluids and tissues [116] to predicting the age of stains and donors [117], transcriptomic analyses provide valuable information for forensic casework. Haas et al. conducted a comprehensive review on the advances in and future prospects of forensic transcriptome analyses in forensic genetics, also addressing the utility of microbial markers for different post-mortem analyses [115].

Although a pipeline for forensic microbiome analysis is still not available [45], the application of NGS-based technologies for microbial post-mortem analysis has the potential to provide valuable insights into the microbial ecology associated with different causes of death and infectious diseases. Further research in this area may help to refine and expand the use of these techniques in microbial post-mortem analysis. Moreover, Oxford Nanopore technology with its reduced machine size and real-time analysis capability is receiving attention from the forensic community [90,94]. In the future, this technology may also be used to study the microbial communities associated with deceased individuals to obtain important forensic information directly on site and in small laboratories.

Alongside nucleic acids, proteins offer valuable insights in forensic microbiology, providing a complementary avenue for obtaining crucial information [118]. Several research groups have devised robust data analysis pipelines for organism identification and taxonomic classification from untargeted liquid chromatography–tandem mass spectrometry (LC-MS/MS) data [119,120], demonstrating the potential of proteomic analysis in microbial forensics. Practical applications include analyzing the microbial proteome of bodily fluids like saliva, enabling the identification of individual habits such as smoking [121]. Furthermore, proteomic analysis can be used to distinguish laboratory-adapted bacteria from closely related wild isolates [118], characterize the growth medium of bacteria [122] and determine the host cells of virus particles [123], highlighting its versatility and potential relevance in forensic investigations.

## 5. Main Applications

### 5.1. The Discriminatory Power of Forensic Microbiology

The microbial composition of an individual is influenced and shaped by numerous factors. These factors encompass a wide range of elements, including genetic predispositions, environmental exposures, dietary habits, lifestyle choices, hygiene practices, medications and geographical location [124,125,126]. Each of these factors plays a significant role in determining the types of microbes that colonize the human body and contribute to the overall composition of the microbiome. As a result, the microbial profile of an individual is highly dynamic and unique, reflecting an individual’s unique life history and offering a unique biological identifier similar to fingerprints [127,128]. 

These unique signatures can be potentially used in forensic studies for individual identification, trace evidence tracking in contact-related crime scenes and geographical localization [53,129,130]. Studies have demonstrated that remnants of human microbiota persist in the environments we inhabit and on the surfaces with which we come into contact [57,103,131,132,133,134,135]. In a 2010 study, Fierer et al. demonstrated that skin-associated bacteria, recoverable from surfaces such as individual computer keys and computer mice, exhibit community structures that enable the discrimination of objects handled by different individuals, even after remaining untouched for up to 2 weeks [57]. Similarly, Meadow et al. investigated the potential of mobile phones to gather data on individuals’ personal microbiome, revealing that a significant portion of bacterial taxa found on participants’ fingers were also present on their own phones. Interestingly, individuals shared more bacterial communities with their own phones than with anyone else’s, suggesting the potential use of microbiome analysis to link individuals with objects on the basis of their microbial composition [133]. 

Similarly, diverse geographical locations harbor unique microbial compositions in their soil and water ecosystems. The analysis of microbial composition from samples collected at crime scenes holds promise in deriving geolocation information [130]. For instance, Jesmok et al. conducted a study employing next-generation sequencing of the bacterial 16S rRNA gene, successfully linking soils to their respective origins with a notable success rate [136]. Likewise, Su et al. employed bacterial genus and composition within the lung tissues of drowning victims to infer the locations of drowning incidents [137]. The comprehensive review by Moitas et al. [130] further underscores the growing importance of forensic microbiology in determining geographical locations, highlighting recent advancements and studies in this emerging field. 

Despite the clear potential of a microbial signature study for forensic investigation, some challenges remain. Unlike traditional identifiers such as DNA or fingerprints, which remain relatively stable throughout a person’s life, the microbiome is dynamic and responsive to various internal and external influences [138]. This instability in microbial composition may be problematic for forensic applications, as it could introduce variability and uncertainty in microbial evidence over time. Moreover, numerous studies have shown that close interactions, such as those within families, between pets and their owners and among coworkers, can lead to similarities in microbial profiles, highlighting the challenge of distinguishing between the microbial signatures of closely related or cohabiting individuals [103,139,140]. Another significant challenge is distinguishing individual differences from the background microbial populations. In a recent study, Simon et al. examined the microbial composition of museum objects, demonstrating that microbial signatures can effectively track exposure to human touch and differentiate objects handled by distinct individuals. Additionally, their research revealed that human contact might overwrite the previous microbial profile of an object, making it difficult to accurately trace individuals based on the microbial signatures left on objects [135].

### 5.2. Sudden Death

Sudden death (SD) can be defined as “a natural, unexpected fatal event occurring within one hour from the onset of symptoms, in an apparently healthy subject, or in one whose disease was not severe to predict such an abrupt outcome” [141].

SD can occur in people of all ages, including infants, children and adults. Forensic medicine encounters a wide range of deaths that are the result of natural causes. Despite advances in diagnosing and treating infectious diseases, a significant number of sudden and unforeseen deaths are attributed to infections [142].

Sudden death due to infectious pathogens is classified in consideration of both the specific organ or general system affected (e.g., myocarditis, meningitis, sepsis, septic shock) and the presumed etiological agent (e.g., bacteria, fungi, viruses, parasites). This is identified with the use of methods of rapid molecular or cultural analyses. The microbiological demonstration of a microorganism as the causative source of sudden infectious death should hopefully be supported by pathological autopsy investigations. The combination of microbiological and pathological evidence is very important to strengthen the hypothesis between cause and effect and limit the possibility of false diagnoses [142]. An important research area in SD is the sudden infant death syndrome (SIDS). A panel of experts that met in San Diego in 2004 proposed a general definition for SIDS as “the sudden unexpected death of an infant <1 year of age, with onset of the fatal episode apparently occurring during sleep, that remains unexplained after a thorough investigation, including performance of a complete autopsy and review of the circumstances of death and the clinical history” [143]. SUDI, or sudden unexpected death in infancy, is a proposed term referring to all sudden and unexpected infant deaths and not just to SIDS. In addition to histopathology and toxicology, ancillary analyses encompass post-mortem microbiology, involving both bacteriology and molecular analyses, conducted on samples obtained during autopsy. These investigations demonstrate that SD in infancy may be caused by various bacteria and viruses in up to 10% of cases. Indeed, bacteria and viruses are found in the blood and tissue samples of infants who die suddenly, as suggested by Prtak et al. [144]. Clinical and forensic assessment, including microbiology, is necessary to properly evaluate the case in such scenarios. A study by Alvarez-Lafuente et al. [145] found that in positive cases, herpes viruses CMV, EBV and HHV-6 could be linked to certain SD cases in infancy. Other implicated pathogens include human metapneumovirus, *N. meningitidis*, *Haemophilus influenzae*, *Streptococcus pneumoniae* and *Bordetella pertussis* [53]. Those SUDI cases, where the presence of a pathogen can explain the cause of death as due to infection, are eventually classified as explained deaths and should be excluded from the SIDS definition. Additionally, research by Leong et al. [146] suggested that a disturbance in gut microbiota, as well as co-infections with *Clostridioides difficile*, *Clostridium perfringens*, *Clostridium innocuum* and *Bacteroides thetaiotaomicron*, may help explain some cases of SIDS.

One of the most important microbial causes of SD both in children and in adults worldwide are *N. meningitidis* infections that can turn into invasive meningococcal disease (IMD) [147]. In Italy, in the last 10 years, IMD was at the center of legal actions and trials. In particular, in 2015/2016, serogroup C meningococci of the clonal complex CC11 (MenC/cc11) was responsible for an epidemic in the Tuscany region [148]. Between December 2019 and January 2020, an outbreak of six cases caused by the same clonal complex MenC/cc11, closely related to the Tuscany strain, was identified in a limited area in the Lombardy region (Brescia and Bergamo counties [149]. All these six cases presented a severe clinical picture, and two of them were fatal. In these two cases, patients—26- and 46-year-old women—were diagnosed with meningococcal sepsis (Waterhouse–Friderichsen) and autopsies were performed because of legal involvement. Diagnosis was confirmed from ante-mortem and post-mortem samples: in one case, *N. meningitidis* serogroup C was isolated in culture from the cerebrospinal fluid; in the other case, whole blood was collected ante-mortem and immediately sent to the Reference Center for Invasive Disease (Policlinico of Milan), for molecular detection and identification. The rapid identification and characterization of IMD cases and an extensive public vaccination campaign contributed to the successful control of this outbreak, caused by a hyper-invasive meningococcal strain [149].

### 5.3. Forensic Microbiology and the Deciphering of the Cause of Infectious Death in Outbreaks

At present, the progress made in obtaining highly accurate microbial fingerprints opens new application potential such as helping to find out the cause of an infectious death. One significant example is the investigation of an infectious outbreak in a hospital. Hospital outbreaks are undesirable and become even more so in the case of fatal events. If death is perceived as an avoidable event for which insufficient preparations have completed or even attributable to third-party assistance, it can lead to litigation. When outbreaks and fatalities occur in neonatal care units, they often lead to mass media and public attention, making investigation a delicate matter. 

Medico-legal disputes evaluate three parameters: possible negligence in the health care, any damage caused and any causal link. The hospital or the primary care center must therefore demonstrate that all appropriate measures have been taken on the basis of current scientific knowledge, not only with regard to the treatment but also concerning diagnostic and care procedures to prove that the correct behavior was adopted, and that the infection was caused by an unforeseeable circumstance.

In these cases, forensic microbiology may be of use to demonstrate whether the hospital measures were correct in terms of prevention, control and therapeutic treatment (traceability, guidelines, operational protocols) proving the ineluctability of the event [53].

For outbreak investigations, it is necessary to place the infectious agent into a more discriminatory category than species, to establish links between cases and sources. The success of a reliable and robust tracking system relies on the quantity and quality of the data collected on isolate identification, location of possible sources, definition of possible reservoirs and determination of transmission routes or evolution dynamics [150,151].

DNA sequence-based typing methods have substituted serological typing offering a fine-tuned bacterial fingerprint that is essential for outbreak investigation. From the 6–10 alleles considered in the classical MultiLocus Sequence Typing (MLST), the typing schemes have extended to core-genome MLST (cgMLST) or whole-genome MLST (wgMLST); however, they do not yet include all potential allelic variations and cover only a limited number of microbial species.

An in-depth discrimination between pathogens at one nucleotide resolution (single nucleotide polymorphisms, SNPs) has been reached by the use of whole-genome sequencing (WGS). WGS also shortens the response time between sample collection and results, boosting surveillance and clarifying outbreaks in a real-time manner [53,152]. For this reason, genomic surveillance is increasingly used to map the spread of microbial pathogens in nosocomial settings because it has the superior power of elucidating microbial radiation, calibrated by time, enabling the detection of complex transmission chains that are not readily apparent from epidemiological data and which can significantly contribute to morbidity and mortality [153].

In the case of nosocomial bacterial outbreaks, WGS data are also used to determine virulence genes, antibiotic resistance determinants, plasmids, prophage, insertion sequences and mobile elements of nosocomial pathogens, making the traceability of emergent clones more efficient [52,154]. 

Genomic data are also used in retrospective studies to detect missed outbreaks or clarify misidentified outbreaks with the aim of improving preparedness and containment measures [155,156]. 

It must be kept in mind that all steps of genomic surveillance (next generation sequencing, bioinformatic analysis, lineage classification, mutation calling, phylogenetic, phylodynamic and phylogeographic inferences) need bioinformatic skills, extensive computational resources, cloud computing platforms, dedicated pipelines designed for specific pathogens, repositories as well as established procedures for the use of WGS data for outbreak detection [157]. Attempts are also in progress to employ artificial intelligence (AI) approaches for the analysis of infectious disease transmission dynamics during and after pathogen outbreaks, and for the development of AI-based models of phylogeny and transmission of public health-relevant pathogens [158]. 

### 5.4. Violence, Violent Death and Forensic Microbiology

Microbiological analysis is increasingly recognized as important in criminal investigations in relation to personal identification, study of trace evidence, sexual violence, determination of individual’s geographic origin and to reconstruct the place and the cause of death. Despite concerns about the correct interpretation and statistical significance, there is a growing awareness of the need to associate microbiological analysis with other post-mortem procedures [41,75,159].

Regarding identification, the research suggests a vast number of unique microbiomes on human skin that can easily spread onto surfaces [65]. Unlike DNA, microbial communities can remain on surfaces for extended periods due to their ability to withstand environmental stressors such as humidity, temperature or UV radiation. Since the composition of microbial communities is specific to each person, it is possible to demonstrate the presence of an individual in a particular location, such as a crime scene, and also the type of traces left, whether it comes from the skin or other body sites [64,160]. However, as the potential forensic value of microbiota traces is not static, their identifying characteristics are lost over time [161]. Variations in the microbiota have been described depending on an individual’s sex and age, but there is still a core microbiome that allows discrimination of each body fluid or body site [161,162].

Microbial markers can differentiate specific bacterial species at different body sites [163,164]. Species that represent a minor component of the body-associated microbial communities tend to be unique and may play a crucial role in personal identification [165]. 

The microbiome composition of the skin varies depending on the body region, including skin thickness, folds, hair follicle density and the presence of eccrine and apocrine glands [53]. Moreover, the microbiota can transfer directly from one individual to another through a handshake, even in the presence of intermediate substrates [166].

Currently, microbial markers of interest can differentiate specific bacterial species found on various bodily sites/fluids, including the skin, vagina, feces, semen, saliva and hair matrix [163]. For example, Leake et al. [167] showed that the salivary microbiota differs between individuals. As for the skin, *Micrococcus* and *Staphylococcus* are the most commonly detected species on the hands, with *Cutibacterium* (previously *Propionibacterium*) species having forensic relevance [160,164]. Wilkins et al. [161] showed that it is possible to recover handprints with a microbiome resembling that found on objects typically used by the subject. 

Concerning sexual violence, the presence of vaginal secretions can be distinguished from other bodily fluids by analyzing the microbial community found in the vagina, which mainly consists of *Lactobacillus crispatus* and *Lactobacillus gasseri* species [168,169]. Díez Lòpez et al. [170] showed that it is possible to differentiate between menstrual, venous, nasal or skin wound blood found at crime scenes, based on microbial presence. In particular, their study primarily utilized massively parallel microbiome sequencing to analyze the unique microbial signatures of different body sites allowing for the classification of fluids based on the distinct microbial populations they contain through microbiome sequencing. This method, using de novo generated 16S rRNA gene sequencing data, employed a taxonomy-independent deep neural network, which was trained on a large dataset of microbiome sequences from various body sites, to accurately predict the origin of a blood sample at a crime scene. However, there are several limitations and challenges associated with microbial analysis in forensic settings. One major challenge is the complexity of mixed samples, which may contain multiple types of body fluids or come from multiple individuals. This can complicate the interpretation of microbial data and lead to potential misclassification. Additionally, the condition of samples (e.g., age, exposure to environmental factors) can affect the stability and detectability of microbial signatures. Another limitation is the reliance on high-quality DNA samples for accurate sequencing and analysis. Degraded or contaminated samples may yield incomplete or misleading microbial profiles. Furthermore, the variability in microbial communities among individuals due to genetic, dietary and environmental factors can introduce uncertainty into the analysis. Normal semen contains bacteria from the genera *Lactobacillus* and *Staphylococcus* like those found on the skin and in the vagina [171]. Some of the stable genera detected in the skin of the penis over time include *Staphylococcus*, *Anaerococcus* and *Prevotella*, with the latter apparently being exclusive to uncircumcised males [172]. The identification of body-site-specific microbial biomarkers by NGS coupled with machine learning is a promising tool to be used in the forensic field [173].

Hair matrices, particularly pubic hair, are a useful source of microbial information, with pubic hair being more stable than hair affected by environmental factors and length [174]. The microbial community of pubic hair was observed to be shared by two individuals who had sexual intercourse, and in a simulation of sexual assault, the aggressor’s microbial profile could be reconstructed, differentiating between hair and pubic hair [174,175].

Various studies have shown that the bacterial species present in deceased individuals (such as *Helicobacter pylori*) or adult hair matrices in different cities can be used to predict an individual’s geolocation, due to climate, rainfall, altitude, soil and energy sources in the environment [176]. 

The analysis of microbial communities present in soil can provide important information about the location of death. Soil communities are also relevant in determining surface or underground death [177]. In burial soil samples, Proteobacteria are the most abundant phylum, while Acidobacteria have decreased and Firmicutes have increased in surface corpse microbial communities [178].

Microbiology in forensic pathology can play a role in identifying microbial markers in cases of death by drowning. The studies have shown that it is possible to distinguish specific bacterial colonization in the blood of drowning victims that is not found in corpses immersed in water [179]. The possibility of discriminating whether the drowning occurred in freshwater or saltwater based on microbial development allows for the identification of the place of drowning and distinguishing between primary and secondary crime scenes [180]. The “diatom test” stands as a gold standard in confirming drowning deaths, regularly utilized by forensic scientists to examine diatoms infiltrating the organs [181,182]. Additionally, the identification of *Aeromonas*, *Pseudomonas* and *Shewanella* species in blood, bone marrow and lungs has proven to be a valuable indicator for diagnosing freshwater drowning [183]. In contrast, seawater drowning is associated with the presence of *Vibrio*, *Photobacterium*, *Listonella*, *Marinomonas* and *Pseudoalteromonas* [180].

Despite all the promising developments, before the successful integration of microbiome-based analyses into forensic casework, two primary challenges must be tackled: conducting a comprehensive evaluation of the method’s strengths and limitations, and establishing a standardized operating procedure that should include laboratory and bioinformatic workflow recommendations [173,184]. 

## 6. Forensic Microbiology at the Border of Paleomicrobiology

Paleomicrobiology is the study of the ancient microbes associated with archaeological materials, such as bones, teeth, coprolites and other organic remains [185]. It uses techniques from microbiology, molecular biology and bioinformatics to identify and characterize the microorganisms that lived in the past. One of the primary goals of paleomicrobiology is to reconstruct the evolution and diversity of microorganisms, as well as to understand the impact that microorganisms have had on human and animal populations [185]. Some of the applications of paleomicrobiology include the identification of ancient pathogens responsible for historical epidemics [186,187], the investigation of the microbial ecology of ancient environments [188], and the study of the coevolution of microorganisms and their hosts over time [114]. Moreover, by studying the microbiomes of ancient humans and comparing them to those of modern humans, paleomicrobiologists have been able to identify changes in the microbiome over time, as well as the factors that have influenced these changes [189,190].

The starting samples for paleo-microbiological analysis are often found in poor conditions, similar to what happens for forensics samples [2]. Thus, the two fields must face some common problems, such as the high level of degradation of the samples and contamination, which deeply affect the quantity and quality of the molecules of interest [114].

The degradation of organic molecules over time, such as DNA and proteins, is a well-known phenomenon that can affect the quality and quantity of molecules in biological samples [191,192].

DNA can be damaged by a variety of factors, including environmental stressors, such as exposure to UV radiation, high temperatures, and chemicals. However, even under optimal storage conditions, DNA can still degrade over time due to natural processes, e.g., hydrolysis, oxidative nucleotide modification, the action of nucleases and other hydrolytic enzymes [193]. Degradation usually leads to the fragmentation of the DNA molecule. As a result, the length of DNA fragments in a sample can decrease over time, making it more difficult to perform accurate analysis [194,195]. In order to, at least partially, overcome this problem, protocols of DNA reparation have been developed and applied to paleo-microbiological analysis [196,197,198]. These protocols allow the nucleotides to be removed that have been “damaged” (e.g., abasic sites and uracil substitutions [196]) and as such improve the quality of the DNA fragments and ultimately increase the robustness and efficiency of ancient DNA analysis [196,197].

Sometimes, fragmentation and decomposition are so extensive that the analysis of ancient DNA is impossible, while proteins are naturally more resistant and can be detected even after millions of years [199,200]. Thus, proteins and paleo-proteomic analysis can provide important information particularly in conditions of extensive DNA degradation [199].

In addition, the type of biological sample can also affect the rate of DNA and protein degradation [201]. Molecules extracted from soft tissues, such as the liver or brain, may degrade faster than molecules from harder tissues, such as bones or teeth, which are more resistant to environmental stressors [202,203].

Another important problem to consider during the microbial analysis of ancient samples is exogenous contamination. Contaminants can come from any step of the analysis, from the burial site to the transport and laboratory handling [204]. To avoid contamination, paleo-microbiological studies must be performed in dedicated clean laboratories, ideally exclusively dedicated to work on ancient remains and by following specific protocols to ensure the authenticity of the results [185,205]. Moreover, bioinformatic tools, like MapDamage 2.0 [206] and AuthentiCT [207], are able to determine the level of DNA degradation in the sample and may be used to distinguish between modern and ancient DNA molecules, since modern DNA molecules are usually less degraded than ancient ones [207].

Despite the challenges that both paleo-microbiological and forensic analyses face, there is potential for these fields to collaborate more closely. For example, to tackle the difficulty of sequencing ancient DNA, specific laboratory procedures have been created with the goal of minimizing contamination and increasing the quantity of extracted molecules from the remains. Thus, protocols and kits designed to work in one field can be re-adapted to work in the other.

## 7. Concluding Remarks

Forensic microbiology represents an emerging discipline with undoubted practical implications. There is a critical need for standardized procedures in this novel field, covering the collection, analysis and interpretation of data. Firmly established protocols would also ensure the validity of the exhibits to be used in criminal prosecutions or civil trials. In this context, the ESGFOR study group has made efforts to standardize sampling in different scenarios related to forensic microbiology [4,5,208]. This is also related to the European Union Council Framework Decision 2009/905/Jha, which offers a legal framework for forensic service providers carrying out laboratory activities. These laboratory procedures should be accredited according to ISO 17025 [209]. All of these forensic service providers should safeguard the chain of custody as described in ISO 21043 [210] of Forensic Sciences, and thus the requirements for the forensic process, focusing on recognition, recording, collection, transport and storage of items of potential forensic value. It includes requirements for the assessment and examination of scenes, but it is also applicable to other activities by the facility that are required to adhere to quality standards. 

For the coming-of-age of forensic microbiology, a rigorous methodological contemplation is needed, achievable only through opinion sharing and discussions within the relevant scientific community, ideally followed by collaboration. This process should encompass the unique contributions of the various disciplines involved, including microbiology, forensic pathology, forensic genetics, infectious diseases and bioinformatics. 

## Figures and Tables

**Figure 1 microorganisms-12-00988-f001:**
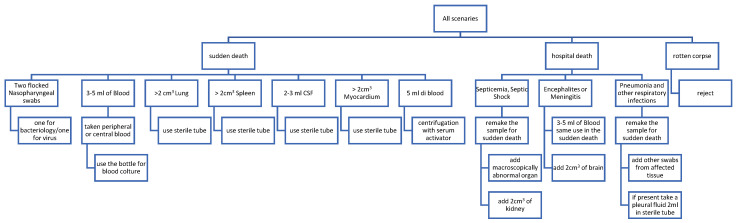
Flow chart of the procedures to be implemented in the event of a death, where it is suspected that the death was caused by a pathogenic microorganism (including viruses). In particular, the flow chart indicates the different types of autopsy samples from which to proceed to search for the infectious agent, and some methodological precautions, for example, some basic information on the tissue samples to be collected and the use of sterile tubes.

**Table 1 microorganisms-12-00988-t001:** Applications of forensic microbiology.

Main Application	Type of Samples	Method	Selected Refs
Determination of post-mortem interval (PMI)	Autopsy tissues	Molecular analyses	[37,46]
Determining the infectious causes of death	Autopsy tissues/Body fluids	Bacteriological, histological and molecular analyses	[14,47]
Determining the type of biological fluid	Swabs or samples of the biological trace	Molecular analyses	[48,49]
Differential diagnosis in shaken baby and child injuries	Autopsy tissues	Bacteriological analyses with histological examination and molecular analyses	[50,51]
Identifying deaths attributable to infectious diseases during outbreaks	Environmental/autopsy tissue/body fluids	Bacteriological and molecular analyses with bioinformatics	[52,53]
Identifying instances of medical malpractice and nosocomial infections	Swabs of specific body regions/body fluid/objects	Bacteriological and molecular analyses	[53,54]
Identifying soil types and specific locations	Soil/objects containing soil	Molecular analyses and comparing the results with the forensic microbiome database (FMD)	[55,56]
Identifying the touch residue left on objects by skin	Swabs on objects	Molecular analyses	[35,57]
Personal identification	Swabs of specific body regions	Molecular analyses	[58,59]
Sexual violence investigations	Swabs of specific body regions/objects	Bacteriological and molecular analyses with comparisons between samples	[60,61]
Post-mortem toxicological analyses	Autopsy tissues	Bacteriological, chromatographic and molecular analyses	[62,63]
Violent death investigations	Swabs of specific body regions/objects	Bacteriological and molecular analyses with comparisons between samples	[64,65]

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
