# Peer review of "Forensic Microbiology: When, Where and How"

_microorganisms, 2024, doi:10.3390/microorganisms12050988_

Round 1

Reviewer 1 Report

Comments and Suggestions for Authors

Dear Editor,

The present review article describes the developments in the field of forensic microbiology. The manuscript is well written and easily readable. On the other hand, in the last year three other reviews regarding forensic microbiology have been published, hence it appear that this manuscript does not provide a noteworthy update of the current literature. A point that should be better described and that would be valuable for the reader is to put more emphasis in the need for standardization of the procedures, with reference to the acrtivity of the ESGFOR group. Being all the coauthors from Europe they should mention EU Council Framework Decision 2009/905/Jha that provides the legal framework for forensic service providers carrying out laboratory activities, which procedurese should be accredited according to ISO 17025. This could be integrate by referring to ISO 21043 Forensic Sciences, not fully developed, that is the pillar for transnational acceptance of laboratory results for forensic purposes.

Minor changes:

The keyword "violent death" sounds not fully appropriate to the objectives of the manuscript, the term "microbiome" could be suitable as a keyword.

Ln 387. In the sentence “…isolated from culture media” it is unclear the type of sample (tissue/organ) collected for bacteriological culture. In the referred article it is not reported.

Reviewer 2 Report

Comments and Suggestions for Authors

This review article provides a valuable overview of the emerging field of forensic microbiology, particularly its applications in identifying the cause of death and investigating outbreaks. However, to enhance its comprehensiveness and impact, I recommend addressing the following points:

Specific Comments:

1. Broaden the Scope:

While the review thoroughly covers certain areas, it could benefit from a more comprehensive discussion of the potential applications of forensic microbiology in analyzing various types of forensic material evidence. This could include:

Individual identification: Discuss the potential of using individual-specific microbial signatures for identification purposes, addressing the challenges associated with distinguishing individual variation from background microbial populations and the need for robust statistical methods.

Tissue source determination: Explore how microbial analysis can help determine the origin of biological materials found at crime scenes, such as differentiating between blood, saliva, semen, or other bodily fluids. Discuss the limitations and challenges associated with this type of analysis, particularly in the context of mixed samples.

Trace evidence analysis: Discuss how the analysis of microbial communities on objects or surfaces can provide insights into their contact with individuals or environments. Address the challenges associated with transfer, persistence, and contamination of microbial traces.

2. Address Limitations and Challenges:

While highlighting the potential of various applications, it's crucial to critically discuss the limitations and challenges associated with using microbiome data in forensic investigations. This includes:

PMI estimation: Discuss the limitations of using the thanatomicrobiome to estimate PMI, considering factors like environmental variability and individual differences that affect accuracy.

Contamination: Expand on specific strategies and techniques for minimizing contamination during post-mortem sampling and data analysis.

Interpretation of results: Provide more specific criteria for differentiating between actual infection and contamination, potentially suggesting specific microbial signatures or abundance thresholds.

Bioinformatic challenges: Discuss the bioinformatic challenges associated with analyzing NGS data in the context of forensic microbiology, addressing issues like contamination, low biomass, and degraded DNA.

Attributing cause of death: Discuss the challenges associated with definitively attributing a sudden death to a specific pathogen.

Microbiome-based evidence: Critically discuss the limitations and challenges associated with using microbiome data in criminal investigations, particularly regarding transfer, persistence, and environmental contamination of microbial traces.

3. Additional Points:

Consider including a table or figure summarizing the main applications of forensic microbiology and the associated methodologies.

Ensure that all references are accurate and up-to-date.

Discuss the ethical and legal considerations associated with using microbiome data in forensic investigations, including privacy, informed consent, and potential bias.

Reviewer 3 Report

Comments and Suggestions for Authors

The submitted manuscript is a review paper aimed at different aspects and applications of forensic microbiology. It would serve as a feature paper opening the special issue focused on the progress of postmortem microbiology research. However, this contribution seems to be the only article submitted to this special issue of Microorganisms, as the manuscript submission deadline has long since passed (30 June 2023).

This narrative report does not stand out from other similar publications, but it is a solid overview of the contemporary state and possible applications of the discussed discipline. Compared to other review reports in this field, it lacks tables and illustrative content that would guide the reader and facilitate reading (e.g. infographics or at least bullet points).

The title of the article aims to answer the questions of when, where, and how post-mortem microbiology would facilitate its application in forensics. The authors defined the applications of forensic microbiology in 4 main areas:

5.1. Sudden death

5.2. Forensic microbiology and the deciphering of the cause of infectious death in outbreaks

5.3. Forensic microbiology at the border of paleomicrobiology

5.4. Violence, violent death, and forensic microbiology

This division seems controversial because it significantly limits the area of post-mortem microbiology by not taking into account several issues such as PMI estimation (including postmortem submersion interval), determining geographical location based on microbial community profiles, personal and taxonomic identification (including sex determination, animal microbiomes), pathogen transmission tracking in undiagnosed infections/sepsis (medical error assessment outside infectious outbreak context), trace evidence tracking in contact-related crime scenes, etc.

Moreover, I have no idea why paleomicrobiology was included as a part of the review paper devoted to forensic microbiology. Paleomicrobiology is generally not ‘forensic’ because it is not connected with litigation issues for court purposes. The term ‘forensic’ generally refers to crime detection and application of scientific techniques for litigious purposes. Adjusting the new title (or changing the title) of the manuscript should resolve discrepancy. In this regard, the introductory statement is also misleading (lines 61-62): “In addition, paleomicrobiology could be regarded as a sister field of forensic microbiology [2]” because ref. no. 2 does not refer to paleomicrobiology.

When discussing the question "when?" no indications for microbiological tests were presented, e.g. the usefulness of necrochemical examinations preceding genetic tests in post-mortem diagnosis of infections and sepsis.

I also have doubts about why the post-mortem microbiological analysis section (the "how?" problem) was limited to the NGS technique only. Besides classical culture-dependent methods, there are other approaches than genetic profiling of the thanatomicrobiome like transcriptomics and proteomics and other technologies (like MALDI-TOF spectrometry, LC/MS microbial proteomics, hybridization techniques, micro-arrays) alternative to NGS/SNP/WGS. At a minimum, the discussion should indicate why they were omitted from this review.

Other issues:

- some elements mentioned in the summary do not appear anywhere in the main text, e.g. immunosuppressive treatments, prosthetic installations;

- lines 123-124: “Women with more subcutaneous adipose tissue decay faster due to increased water content.“ – the water content is generally lower in women than men (due to a higher percentage of fat) which is included for instance in the Widmark formula for calculating blood alcohol content;

- lines 346-247: “Sudden death (SD) refers to an unexpected and rapid death that occurs within a short period, typically within one hour of the onset of symptoms” – this is not an exhaustive definition of sudden death.

Reviewer 4 Report

Comments and Suggestions for Authors

The manuscript is well-organized overall and provides much information to researchers in related fields.

However, there should be a more detailed discussion on the change of microbiota in terms of chain of custody. How does the microorganisms in a forensic evidence change over time? What are the changes caused by contamination and how can they be managed?

Round 2

Reviewer 2 Report

Comments and Suggestions for Authors

This manuscript has been revised adequately and now appears acceptable for publication. The reviewer appreciates the authors' positive responses again.